# A conserved enzymatic toolkit targeting host cell metabolism is associated with *Cryptococcus neoformans* intracellular survival in protozoal and mammalian phagocytic cells

Quigly Dragotakes *, Ella Jacobs, Gracen Gerbig, Seth Greengo, Anne Jedlicka, Amanda Dziedzic, Arturo Casadevall

Department of Microbiology and Microbiology and Immunology, Johns Hopkins Bloomberg School of Public Health, Baltimore, Maryland, United States of America

* qdragot@jhu.edu

## Abstract

The outcome of the interaction between *Cryptococcus neoformans* and infected hosts can be determined by whether the fungal cell survives ingestion by phagocytic cells. This applies to both unicellular and multicellular hosts such as amoeba and animals, respectively. Ingestion by phagocytic cells results in the formation of the cryptococcal phagosome but this structure has proved difficult to isolate. In this study, we report the successful isolation of cryptococcal phagosomes from murine and human phagocytes, followed by their characterization using proteomic and transcriptional analysis. Comparison of cryptococcal proteins from *Acanthamoeba castellanii*, *Mus musculus*, and *Homo sapiens* phagocytes revealed the existence of a shared set suggesting a conserved fungal response to ingestion by phagocytic cells. Given that the cryptococcal intracellular pathogenic strategy is ancient, dating to at least to the cretaceous epoch, these results are consistent with the notion that the fungal response to ingestion reflects the result of selection pressures by environmental ameboid predators over eons of evolutionary time. We propose the existence of a conserved cryptococcal toolkit for intracellular survival that includes metabolic enzymes, which disrupt host cell metabolic function, thus providing a common strategy for cryptococcal survival after ingestion by phylogenetically distant phagocytic hosts.

## Author summary

A fundamental problem in the field of microbial pathogenesis is the mechanism by which microbes can cause disease in diverse hosts. In contrast to specialized pathogenic microbes that have an association with single or few hosts, generalist-type pathogens must survive in diverse hosts and thus must have

**Data availability statement:** All sequence files and sample information have been deposited at NCBI Sequence Read Archive, NCBI BioProject: PRJNA1270087, and resulting proteomic and transcriptomics data are included in the supporting information.

**Funding:** This work was supported by the National Institutes of Health (AI33774 to AC; AI33142 to AC; and HL-59842-01 to AC). The funders had no role in study design, data collection and analysis, decision to publish, or preparation of the manuscript.

**Competing interests:** The authors have declared that no competing interests exist.

non-specialized strategies to subvert cellular defenses, such as phagocytic cells. In this study we analyzed the interaction of the human pathogenic fungus Cryptococcus neoformans with amoeba and mouse and human macrophages, each phagocytic cell, comparing fungal proteins released into these cells after ingestion as well as transcriptional changes. C. neoformans is a generalist-type pathogenic fungus capable of causing diseases in mammals, fish, reptiles and insects. Despite the phylogenetic differences between these three types of phagocytic cells, their interaction with C. neoformans was remarkably similar. The results are consistent with the hypothesis that for C. neoformans the capacity for mammalian virulence emerged accidentally from selection pressures driven by phagocytic predators in soils such as amoeba. Furthermore, the results imply the existence of a conserved cryptococcal toolkit in the form of released proteins that enhance intracellular fungal cell survival by altering host phagocytic cell metabolism.

## Introduction

*Cryptococcus neoformans* is an ancient environmental fungus with an intracellular pathogenic strategy dating back to the cretaceous epoch [1]. *C. neoformans* has trans-kingdom pathogenic potential as evident from its capacity to cause disease in plants and diverse animals such as insects, fish, reptiles, and mammals [1]. Despite its life cycle not requiring animal infection, *C. neoformans* is a human pathogen responsible for over 100,000 deaths annually, mostly in immune compromised populations, and is a significant global public health burden [2]. The World Health Organization (WHO) recently categorized *C. neoformans* as a critical priority pathogen [2].

Amoeboid predation of environmental fungal pathogens is hypothesized to have selected for traits that also allow for survival in mammalian macrophages, and thus function as virulence factors [3]. As evidence of this, repeatedly exposing subsequent generations of *C. neoformans* to amoeba predation resulted in variants with increased macrophage toxicity [4]. Furthermore, comparison of the interaction of distantly related cryptococcal strains with macrophages found conservation of cryptococcal pathogenic intracellular strategy over time and speciation [5]. Comparison of intracellular phenotypes of *C. neoformans* and *C. gattii* revealed a striking similarity in virulence factors, transcriptional profile, etc., despite divergence over 50 million years ago [2]. Similarly, clinical isolates harvested before, during, and after human and cockatoo infections revealed stability in *C. neoformans* genomics, virulence factor expression, and overall virulence [6]. This combination of strong conservation of intracellular response in mammalian infections and virulence in amoeba infections fits with the notion that adaptations gained from interactions with amoeba are relevant for mammalian macrophage infections.

Immunologically intact mammals are remarkably resistant to invasive mycoses since the combination of elevated basal temperatures combined with adaptive immunity contains or eliminates most infections without progression to disease [7].

Cryptococcal infections are similarly rare in non-immunocompromised individuals. However, this rarity could change as global temperatures increase and select for heat tolerant fungal strains. New fungal pathogens may emerge that are similarly conditioned to survive in human phagocytic cells due to adaptations made to survive environmental ameboid predators [8]. Consequently, it is extremely important to understand the precise molecular pathways *C. neoformans* employs for survival in phagocytic cells.

The extent to which *C. neoformans* cells manipulate host defenses to achieve long-term phagosome survival is an area of active research. Secretion of immune modulatory microbial proteins is documented in *C. neoformans* and other microbes [9–14]. *C. neoformans* proteins may enter host cells by several mechanisms: proteins associated with polysaccharide secretion, proteins released with extracellular vesicles (EVs) and proteins released during degradation of fungal cells [10,15]. Previous characterization of the cryptococcal proteins secreted during infection was performed on whole cell lysate [9,16]. While this method returned diverse proteins that undoubtedly influence infection outcomes, proteins from multiple routes of dissemination can be expected to be present. Consequently, we are interested in characterizing the proteins secreted directly into the cryptococcal phagosome, as these proteins could play a role in phagosome maturation and non-lytic macrophage-to-macrophage (Dragotcytosis) transfer events.

Historically, it was difficult to isolate and study phagosomes containing *C. neoformans* due to a variety of factors [9]. Continual intracellular replication and capsule growth substantially increases the total volume of the ingested yeast, physically straining the phagolysosomal membrane, which makes it friable during isolation attempts [15,17]. Cryptococcal phagosomes are much larger than bacterial phagosomes, which makes them vulnerable to disruption by shear forces. In addition, secreted phospholipases further compromise the integrity of the phagolysosomal membrane. The result is a particularly fragile phagosome which easily lyses during most isolation attempts which has hindered the study of this critical structure [9]. Here, we revisited this problem using techniques originally described for the recovery of *Aspergillus* phagosomes and succeeded in recovering the cryptococcal phagosome [18]. The protocol was easily applied to *C. neoformans* with no major changes. We then employed proteomic and transcriptomic analysis to identify the *C. neoformans* response to ingestion by and infection of amoebas as well as murine and human phagocytes and identified several sets of proteins conserved across species and polarization states, as well as those which differ. The resulting list of proteins and genes identified and expressed at similar levels during infection across all three species are likely to constitute a toolkit for cryptococcal intracellular pathogenesis. Conversely, the suite of effector proteins secreted uniquely in mammalian infections provides insight into what makes *C. neoformans* such a generalist mammalian pathogen and may lead to the identification of targets for new therapeutic interventions.

## Methods

### Ethics statement

Murine experiments were carried out in accordance with protocol MO21H124 approved by The Institutional Animal Care and Use Committee. Mice were sacrificed using $CO_2$ asphyxiation. Human samples did not require IRB approval, only anonymized PBMC samples were provided in the form of Leukopaks.

### Cryptococcus neoformans

*C. neoformans* strain H99 was originally obtained from the laboratory of John Perfect (Durham, NC) and *C. neoformans* strain H99_GFP from the Robin May lab [19]. Stock cultures were stored at -80 ºC long term and then streaked onto yeast peptone dextrose (YPD) agar plates prior to growth for experimental work. Liquid cultures were seeded from solid media colonies into YPD broth and cultured at 30 ºC with 180 rpm rotation. Knockout mutants were acquired from the Madhani Laboratory, generated from *C. neoformans* strain KN99α as previously described [20].

## Acanthamoeba castellanii

*Acanthamoeba castellanii* ATCC 30234 cells were maintained in 6 cm petri dishes with 6 mL peptone yeast extract glucose (PYG) (20 g Bacto peptone, 1 g yeast extract, 0.4 mM $CaCl_2$ • $2H_2O$, 4 mM $MgSO_4$ • $7H_2O$, 2.5 mM $Na_2HPO_4$ • $7H_2O$, 2.5 mM $KH_2PO_4$, 0.05 mM Fe $(NH_4)_2(SO_4)_2$ • $6H_2O$, 0.1% sodium citrate dihydrate, 100 mM glucose) pH 6.5 at 25 °C for 48 h. Amoeba were gently lifted from the petri dish with a cell scraper and transferred to a microcentrifuge tube, then centrifuged at 400 *g* for 5 min. Supernatant was removed and cells were resuspended in fresh PYG. Amoeba were counted with a hemacytometer and diluted, then $10^6$ amoeba/well were seeded into a 6-well tissue culture plate and left to adhere for 1 h at room temperature. The plate was covered to protect it from light. After cells had adhered, media was removed by gently pipetting and replaced with 500 µL 100 mM $CaCl_2$, 400 mM $MgSO_4$ • $7H_2O$, 250 mM $Na_2HPO_4$ • 7H2O, 250 mM $KH_2PO_4$, 0.1% sodium citrate dihydrate, 5 mM Fe $(NH_4)_2(SO_4)_2$ • $6H_2O$. Cryptococcal cultures were diluted in Ac buffer and added at an MOI of 1, then incubated at 25 °C in darkness for 2 h. After incubation, cultures were gently pipetted to remove adhered cells and transferred to a clean microcentrifuge tube. Cells were centrifuged at 400 *g* for 10 min and the pellet was retained. To lyse, the cells were resuspended in 200 µL of 10 mM HEPES supplemented with cOmplete protease inhibitor and incubated on ice for 30 min. Lysis was ensured by pulling the cell suspension 10 times through a 27-gauge needle. Samples were centrifuged at 1,500 *g* for 10 min and the supernatant was kept for proteomic analysis. The pellet was resuspended with 500 µL of Trizol reagent and stored at -80 °C for later use.

## Mammalian phagocytes

Murine bone marrow derived macrophages (BMDMs) were harvested as previously described from 6-week-old C57BL/6 female mice from The Jackson Laboratory [21]. In short, BMDMs were harvested from hind leg bones and were differentiated by seeding 10 cm tissue culture treated dishes in DMEM with 10% FBS, 1% nonessential amino acids, 1% penicillin-streptomycin, 2 mM Glutamax, 1% HEPES buffer, 20% L-929 cell conditioned supernatant, 0.1% beta-mercaptoethanol (BMDM differentiation media) for 6 days at 37 °C and 9.5% $CO_2$. BMDMs were used for experiments within 5 days after differentiation. BMDMs were activated with 0.5 ug/mL LPS and 10 ng/mL Mouse IFN-γ for M1 polarization or 20 ng/mL Mouse IL-4 for M2 polarization for 16 h prior to experiments.

Human peripheral blood monocytes (PBMCs) were isolated from leukopaks following previously established methods. These methods are exempt from IRB approval as we only receive completely anonymized isolated cells from the collection site. In short, human circulating cells were obtained from Leukopaks and separated into approximately 5 mL aliquots in 50 mL centrifugation tubes. We then added 30 mL of RIPA media and an underlay of 10 mL Percol and established a monocell layer via centrifugation at 400 *g* for 30 m. The monocell layers were transferred to fresh tubes and 40 mL fresh RPMI added, then pelleted again at 400 *g* for 5 min. The cell pellets were decanted, then combined and resuspended in 20 mL fresh RPMI. A portion is taken for cell counting before pelleting again at 400 *g* for 5 min. Cells were resuspended according to cell count to comply with Pan Monocyte Isolation Kit manufacturer protocol (Miltenyi Biotec) followed by LS Column purification (Miltenyi Biotec). Cells were then seeded either 3 x $10^7$ in T175 flasks or into tissue culture plates described for following experiments. PBMCs were activated with 0.5 ug/mL LPS and 10 ng/mL Human IFN-γ for M1 polarization or 20 ng/mL Human IL-4 for M2 polarization for 16 h prior to experiments.

## Phagolysosome isolation

Differentiated BMDMs were seeded at a density of 4 x $10^6$ per well of a 4-well plate (21.8 $cm^2$/ well). To compare phagosomal content from M0, M1, and M2 polarized phagocytes 2 wells per condition were activated overnight with LPS (0.5 µg/ mL) and IFN-γ (10 ng/ mL, Roche) for M1 or 20 ng/ mL IL-4 for M2 at 37 °C with 9.5% $CO_2$. Mouse and Human IFN-y and IL-4 were used for their respective species cell lines. BMDMs were infected for two hours with *C. neoformans* strain H99 for proteomics experiments and *C. neoformans* strain H99_GFP for immunofluorescence at an MOI of 2 with 18B7 (10 µg/ mL)

antibody opsonization. Phagocytosis was synchronized by centrifuging infected plates at 500 *g* for 1 min immediately after adding the opsonized yeast. *C. neoformans* containing phagolysosomes were isolated as previously described [18]. Phagosomes from 2 wells per condition were harvested and concentrated into 150 μL PBS per condition.

To achieve coverslip adherence, 20 μL of each concentrated suspension was seeded into each well of a 24-well plate containing 1 mL PBS and Poly-D Lysine coated cover slips. Phagosomes were placed at 4°C and allowed to adhere overnight. Coverslips were fixed with 4% PFA and blocked with 2% BSA for 1 h before staining. H99 GFP containing phagosomes were first stained with a 1:100 concentration of anti-V-ATPase E1 polyclonal antibody (PA5–29899) at 37 °C with orbital shaking. Samples were washed thrice with PBS and stained with Goat Anti-Rabbit IgG H&L conjugated to a Texas Red fluorophore (ab6719). Coverslips were mounted on slides with 2 μL ProLong Gold Antifade.

### Proteomics

Whole cells and isolated phagosomes were lysed by resuspension in 10 mM HEPES buffer and passed through 26 ¾ gauge syringes to ensure mammalian cell lysis. Protein concentrations were determined by BSA and sent for mass spectroscopy protein identification.

Protein extracts were buffer exchanged using SP3 paramagnetic beads (GE Healthcare) [22]. Briefly, protein samples (20 μg) were brought up to 100 μL with 10 mM TEAB + 1% SDS and disulfide bonds reduced with 10 μL of 50 mM dithiothreitol for 1 hour at 60C. Samples were cooled to RT and pH adjusted to ~7.5, followed by alkylation with 10 μL of 100 mM iodoacetamide in the dark at RT for 15 minutes. Next, 100 ug (2 μL of 50 μg/ μL) SP3 beads were added to the samples, followed by 120 μL 100% ethanol. Samples were incubated at RT with shaking for 5 minutes. Following protein binding, beads were washed with 180 μL 80% ethanol three times. Proteins were digested on-bead with 2ng trypsin (Pierce) in 100 μL 25 mM TEAB buffer at 37C overnight. The resulting peptides were separated from the beads using a magnetic tube holder. Supernatants containing peptides were acidified and desalted on u-HLB Oasis plates. Peptides were eluted with 60% acetonitrile/ 0.1% TFA and dried using vacuum centrifugation.

Each of the 12 dried peptide samples were labeled with one of the unique TMTpro 18-plex reagents (Thermo Fisher, Lot WJ330834 First 16 + XA343800 134C + 135) according to the manufacturer's instructions. All TMT labeled peptide samples were combined and dried by vacuum centrifugation.

The combined TMT-labeled peptides were re-constituted to 2 mL in 10 mM TEAB in water and loaded on a XBridge C18 Guard Column (5 μm, 2.1 x 10 mm, Waters) at 250 μL/ min for 8 min prior to fractionation on a XBridge C18 Column (5 μm, 2.1 x 100 mm column (Waters) using a 0–90% acetonitrile in 10 mM TEAB gradient over 85 min at 250 μL/min on an Agilent 1200 series capillary HPLC with a micro-fraction collector. Eighty-four 250 μL fractions were collected and concatenated into 24 fractions according to Wang et al 2011 [23].

TMT labeled peptides in each fraction were analyzed by nanoflow reverse phase chromatography coupled with tandem mass spectrometry (nLCMS/MS) on an Orbitrap-Fusion Lumos mass spectrometer (Thermo Fisher Scientific) interfaced with an EasyLC1000 UPLC. Peptides will be separated on a 15 cm × 75 μm i.d. self-packed fused silica columns with ProntoSIL-120–5-C18 H column 3 μm, 120 Å (BISCHOFF) using an 2–90% acetonitrile gradient over 85 minutes in 0.1% formic acid at 300 nl per min and electrosprayed through a 1 μm emitter tip at 2500 V. Survey scans (MS) of precursor ions were acquired with a 2 second cycle time from 375-1500 m/z at 120,000 resolution at 200 m/z with automatic gain control (AGC) at 4e5 and a 50 ms maximum injection time. Top 15 precursor ions were individually isolated within 0.7 m/z by data dependent monitoring and 15s dynamic exclusion and fragmented using an HCD activation collision energy 39. Fragmentation spectra (MS/MS) were acquired using a 1e5 AGC and 118 ms maximum injection time (IT) at 50,000 resolution.

Fragmentation spectra were processed by Proteome Discoverer (v2.5, ThermoFisher Scientific) and searched with Mascot v.2.8.0 (Matrix Science, London, UK) against RefSeq2021_204 database. Search criteria included trypsin enzyme, two missed cleavage, 5 ppm precursor mass tolerance, 0.01 Da fragment mass tolerance, with TMTpro on N-terminus and

carbamidomethylation on C as fixed modifications and TMTpro on K, deamidation on N or Q as variable modifications. Peptide identifications from the Mascot searches were processed within PD2.5 using Percolator at a 5% False Discovery Rate confidence threshold, based on an auto-concatenated decoy database search. Peptide spectral matches (PSMs) were filtered for Isolation Interference <30%. Relative protein abundances of identified proteins were determined in PD2.5 from the normalized median ratio of TMT reporter ions from the top 30 most abundant proteins identified. ANOVA method was used to calculate the p-values of mean protein ratios for the biological replicates set up using a non-nested (or unpaired) design. Z-score transformation of normalized protein abundances from a quantitative proteomics analysis using isobaric mass tags was applied before performing the hierarchical clustering based on Euclidean distance and complete (furthest neighbors) linkage.

## Transcriptomics

Differentiated BMDMs were seeded into 6-well tissue culture treated plates at a density of $10^6$ cells/ well in media sans L929-conditioned supernatant (BMDM growth media). Phagocytes were activated overnight (16 h) with 0.5 µg/ mL LPS and 10 ng/ mL IFN-γ for M1, 20 ng/ mL IL-4 for M2, or no supplementation for M0. Mouse and Human IFN-y and IL-4 were used for their respective species cell lines. *C neoformans* cultures were opsonized with 18B7 mouse IgG monoclonal antibody for 10 min prior to infection at MOI 1. BMDMs were collected 18 HPI from plates using non-enzymatic cell lifter and pelleted at 500 *g* for 5 min. Cells were then lysed by resuspension in 10 mM HEPES buffer and passed through 26 ¾ gauge syringes to ensure mammalian cell lysis. Yeasts were then pelleted by centrifugation at 2300 *g* for 5 min, resuspended in TriZol reagent, and frozen at -80 ºC for RNA extraction.

Yeasts were thawed in 1 mL of Trizol on ice and homogenized in the FastPrep 24 (MP Bio) with Lysing Matrix C Fast Prep tubes at speed 6 for 30 sec, 4 times. Homogenates were held on ice between each cycle. After homogenization, RNA extraction was performed using the PureLink RNA Mini kit with on-column DNase treatment (ThermoFisher). Quantitation of total RNA was performed with the Qubit RNA HS Assay Kit and Qubit Flex Fluorometer (ThermoFisher), and quality assessment was performed by High Sensitivity RNA ScreenTape analysis on an Agilent TapeStation 4200. RNA-Seq Libraries were prepared using the Universal Plus mRNA-Seq Library prep kit (Tecan Genomics) incorporating unique dual indexes. Libraries were assessed for quality by High Sensitivity D5000 ScreenTape on the 4200 TapeStation (Agilent Technologies). Quantification was performed with NuQuant reagent and by Qubit High Sensitivity dsDNA High Sensitivity Assay Kit, on Qubit 4 and Qubit Flex Fluorometers (Tecan Genomics/ThermoFisher).

Libraries were diluted and an equimolar pool was prepared, according to manufacturer's protocol for appropriate sequencer. An Illumina iSeq Sequencer with iSeq100 i1 reagent V2 300 cycle kit was used for the final quality assessment of the library pool. For deep mRNA sequencing, a 200 cycle (2 x 100 bp) Illumina NovaSeq 6000 S1 run was performed at Johns Hopkins Genomics, Genetic Resources Core Facility, RRID:SCR_018669. RNA-seq data was analyzed with Partek Flow NGS Software as follows: pre-alignment QA/QC; alignment to C. neoformans Reference Index using STAR 2.7.8a; post- alignment QA/QC; quantification of gene counts to annotation model (Partek E/M); filter and normalization of gene counts; identification and comparison of differentially expressed genes with GSA (gene specific analysis). Reference Genome: NCBI: GCF_000149245.1_CNA3. All sequence files and sample information have been deposited at NCBI Sequence Read Archive, NCBI BioProject: PRJNA1270087.

## Statistical analyses

Gene Ontology (GO) analysis was performed within R (4.4.1) using the topGO package (2.56.0 [24]). The full list of annotated genes associated with at least one GO term was used as the background gene universe and significant genes were marked logically according to the comparison described. Enrichment was determined via weight01 algorithm and Fishers exact test. Enriched GO terms were consolidated to parental clusters using rrvgo (1.16.0) [25]. KEGG pathway analysis utilized KEGGREST (1.44.1 [26]) in the same manner and a < 0.05 FDR cutoff was used in both analyses. We pooled upregulated and downregulated gene lists for all analyses.

**Dragotcytosis frequency analysis**

5 x 10⁴ Differentiated BMDMs were seeded in 35 mm dishes with 14 mm inset coverslips (MatTek) and M1 activated as previously outlined. Once activated, the BMDMs were infected with 18B7 (10 µg/ mL) opsonized *C. neoformans* at MOI 1 for 1 h. The dishes were washed with fresh BMDM media to remove any lingering extracellular yeast then imaged every 2 min for 24 h under phase contrast on a Zeiss Axiovert 200M scope with 37 ℃ and 9.5% $CO_2$ incubation. Each infected macrophage was then manually tracked through each frame and the frequency of non-lytic macrophage-to-macrophage transfer events (Dragotcytosis [27]) was calculated as total Dragotcytosis events per total infected macrophages.

## Results

### Cryptococcal proteins released during intracellular residence

We used our recently established protocol to isolate and identify *C. neoformans* proteins secreted into the host cell during amoeba and mammalian macrophage infection from whole cell lysate of phagocytic cells [9] (S9 Table). We identified 254 total *C. neoformans* proteins in mammalian and amoeba phagosomes. The presence of these proteins was consistent between host species with 246 proteins identified in all three samples of all three species. We were able to identify some small differences in the abundance of certain proteins between species, mostly when comparing mammals to amoeba (Fig 1A). We then clustered the identified proteins according to relative abundance (Fig 1B). We performed GO and KEGG analysis on the commonly identified proteins and discovered various biosynthesis and metabolic response ontologies along with metabolic KEGG pathways (S1 Table).

The overall conservation of specific secreted proteins and general consistency of abundance suggests a conserved host cell response to ingestion. Ontology and pathway enrichment analysis of the total list of identified proteins suggests their potential for modifications to metabolic pathways and reorganization of cytoskeletal elements (S1A and S1C Fig and S1 and S3 Tables).

Previously, we reported that M1 cryptococcal phagosomes represented a high stress environment for *C. neoformans* relative to M2 phagosomes [21]. Hence, macrophage polarization states were used to create and compare high and low stress cryptococcal phagolysosomes while maintaining an overall similar environment of phagocytic ingestion. Amoebae are not known to polarize so we focused on murine BMDMs and human PBMCs, either M1 polarized, M2 polarized, or M0 unpolarized. We found most significant differences between species rather than between polarization states, again found impressive conservation with all 709 proteins identified in every sample (Fig 2 and S4 Table). Gene ontology enrichment analysis showed a similar focus on metabolism changes but also displayed unique organelle movement including components of the exocyst complex (S1B and S1D Fig and S5 Table).

### *C. neoformans* phagosomal proteins

We adapted our protocol to isolate phagolysosomes from infected host cells, based on established protocols of *Aspergillus fumigatus* [18]. Gentle cell lysis combined with ATP treatment and ficoll layering allowed isolation of *C. neoformans* containing phagolysosomes from both human and murine. The isolation of phagosomes was confirmed via immunofluorescence, collecting V-ATPase positive yeasts from the phagocytes (S2 Fig). Next, we isolated phagolysosomes from murine BMDMs and human PBMCs of all polarizations (M0, M1, and M2) before repeating this proteomic analysis. Comparative analysis revealed a suite of fungal proteins associated with the cryptococcal phagosome, which we surmise are either embedded into phagosomal membranes or secreted into the phagolysosomal space. We identified 676 total proteins which, again, were all discovered across every sample. The largest differences in cryptococcal protein content were again between species, with 267 proteins showing different abundances when comparing pooled Mouse vs Human samples and only a handful differing between polarization states when comparing pooled M1 vs M2, M2, M1 vs M0, and M2 vs M0 (Fig 3 and S6 Table). We also noted increased separation between mouse and human

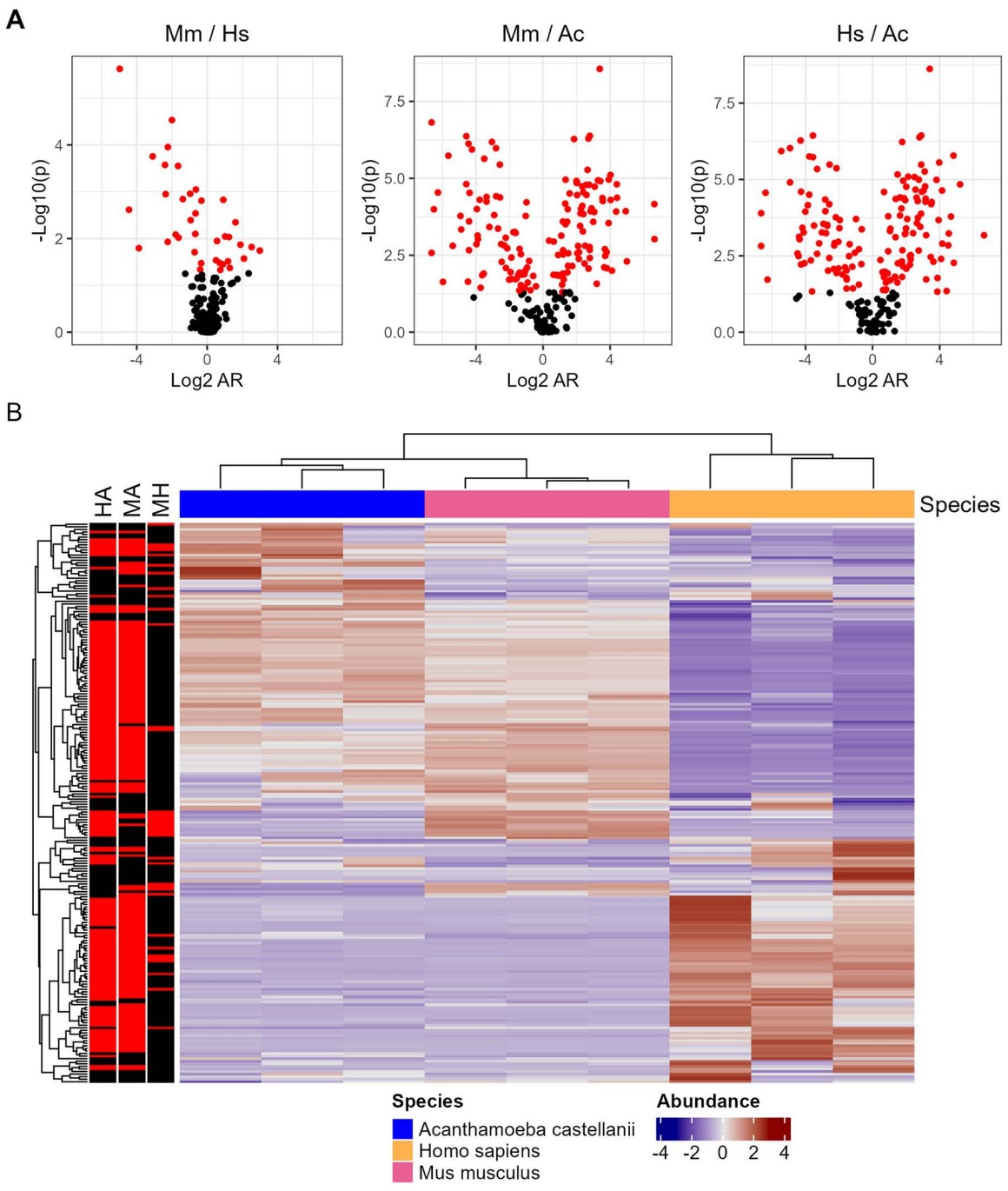

**Fig 1. Proteomic analysis of *C. neoformans* peptides from whole host cell lysate isolated after infection and compiled into quantitative protein data. A.** Volcano plots of individual species comparisons of abundance. (AR: abundance ratio; ratio of calculated abundance of each given protein. Mm: *Mus musculus*. Hs: *Homo sapiens*. Ac: *Acanthamoeba Castellani*) **B.** Abundance comparisons of individual species combinations. Mouse and Human samples correlate well while Amoeba is more unique. Genes are normalized according to row and those which satisfy P < 0.05 are highlighted in red (HA = Human vs Amoeba, MA = Mouse vs Amoeba, MH = Mouse vs Human). Each condition is comprised of three biological replicates, each in technical triplicate.

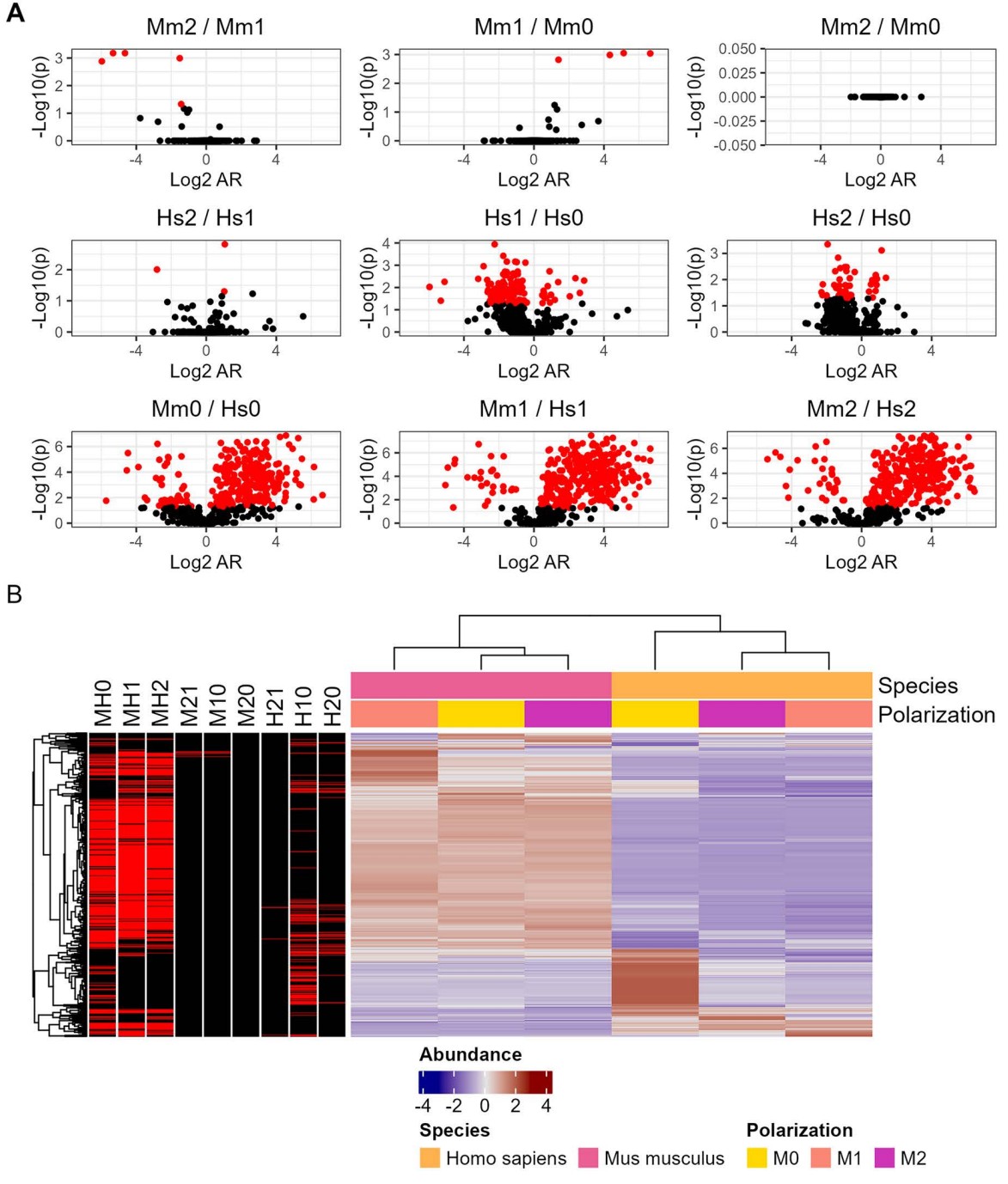

**Fig 2. Proteomic analysis of *C. neoformans* peptides from whole host cell lysate isolated after infection and compiled into quantitative protein data. A.** Volcano plots of murine and human infections across polarization states. **B.** Abundance comparisons of individual conditions. Genes are normalized according to row and those which satisfy P < 0.05 are highlighted in red (MH0 = Mouse M0 vs Human M0, M21 = Mouse M2 vs Mouse M1, etc.). Each condition is comprised of three biological replicates, each in technical triplicate.

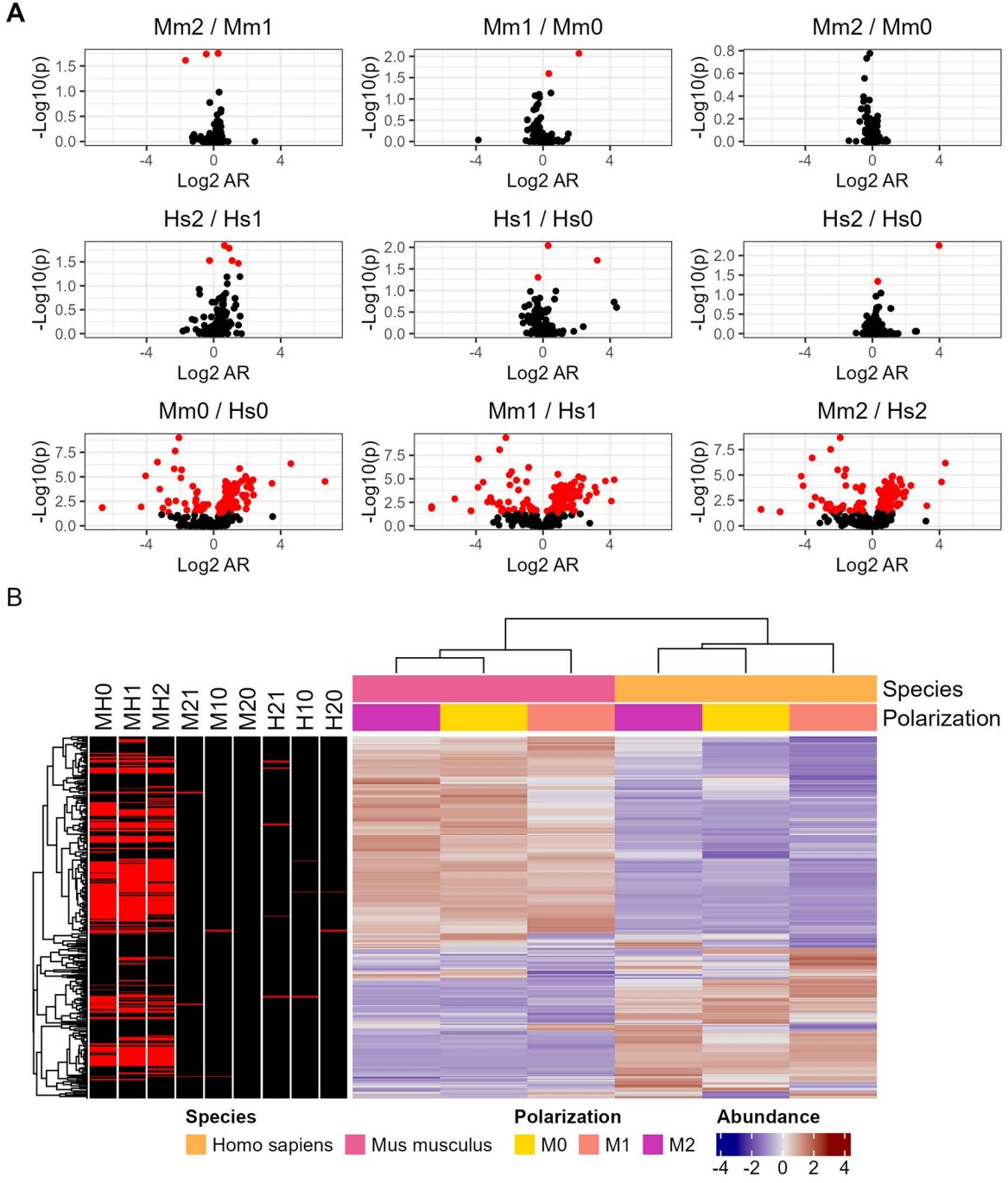

**Fig 3. Proteomic analysis of *C. neoformans* peptides from phagolysosomes isolated after ingestion by mouse and human phagocytes of various polarizations and compiled into quantitative protein data. A.** Volcano plots of individual polarization state and species comparisons. **B.** Abundance comparisons of pooled mouse and human samples. Genes are normalized according to row and those which satisfy P < 0.05 are highlighted in red (MH0 = Mouse M0 vs Human M0, M21 = Mouse M2 vs Mouse M1, etc.). Each condition is comprised of three biological replicates, each in technical triplicate.

samples compared to the prior analysis of proteins in macrophage lysates, even when comparing M1 populations. Gene ontology and pathway enrichment predominantly revealed stress response elements, known phagocytosis and endocytosis associated proteins, and cellular transport-related categories including clathrin binding, microtubule-based processes, myosin complex, and vesicle mediated transport (S3A and S3C Fig and S7 and S8 Tables). As we would expect from isolated phagosomal samples, there was a strong emphasis on phagocytosis and endocytosis pathways, validating the success of our technique.

A subset of proteins was identified in both the isolated phagolysosomes and the whole cell lysates, despite the overall divergence of identified proteins. Ideally, proteins secreted in the phagolysosome would also be present in the whole cell lysate given that cryptococcal phagosomes are leaky as a result of membrane disruption [28]. However, *C. neoformans* also secreted proteins associated with vesicles and EPS into the cytosol. We observed a small portion of 36 overlapping genes, roughly 8–10% of the total proteins discovered in either list (Fig 4A). Abundances were similar across all samples and clustering was mostly by species, isolation method, then polarization, as expected. The notable exception being the Human M0 sample with the lowest abundances across every protein (Fig 4B). Gene ontology and KEGG pathway enrichment shared a similar pattern in which a minority of terms (approximately 18%) were shared, and several unique terms appeared when analyzing the overlapped proteins separately (S3B and S3D Fig and S9 and S10 Tables).

## Transcriptional analysis of ingested cryptococcus neoformans

Understanding the transcriptional response of *C. neoformans* to ingestion can provide context for the proteins identified by proteomics. Thus, we isolated *C. neoformans* after ingestion by amoeba, murine BMDMs, and human PBMCs of different polarizations for RNA sequencing. First, we analyzed the transcriptional profile of *C. neoformans* ingested by macrophage/ monocytes in different polarization states of both mice and human cells compared to stationary phase *C. neoformans* in YPD media at optimal growth conditions. This comparison revealed a range of differentially expressed genes across these samples with most of the significant results either shared across all samples or unique to individual conditions (Fig 5). A total of 359 differentially expressed genes (DEGs) were shared across all samples with similar transcriptional profiles, suggesting a common response to ingestion across species and polarization states (Fig 5 and S11 Table). Several of these 359 genes are already characterized and known to be essential for virulence and capsule generating phenotypes and most correlate to known metabolic functions prioritizing anabolic processes like gluconeogenesis (S4 Fig and S12 Table). Next, we compared M1 and M2 polarized phagocytes in both mouse BMDMs and human PBMCs. We identified more unique DEGs separately than shared between the two species but isolated a list of 53 genes significantly differentially regulated in both (Fig 6A and S13 Table). Unexpectedly, the direction of regulation was only consistent among a subset of these genes (Fig 6B). Additionally, GO and KEGG enrichment analysis did not yield clear effector pathways (S6A and S6C Fig and S14 Table).

We found few differentially regulated genes that overlapped with secreted protein products, especially those with consistent differences in abundances and regulation (e.g., a gene upregulated in Mm M1 vs Hs M1 infection whose protein product is also found in higher abundances in Mm M1 lysate vs Hs M1 lysate). To probe the accuracy of our gene lists, we first identified several hits which are already known to be significantly associated with virulence in the wider literature (Table 1). We then chose several genes with uncharacterized knockouts and subjected them to phenotypic testing, identifying several candidates for investigation which modulated the rate of Dragotcytosis (Table 2 and S5 Fig).

## Host cells proteomics

Except for the phagosome isolation protocol, our experimental design allowed us to purify proteins and material from the amoeba, macrophages, and monocytes as well. We included host cell whole cell lysate in our proteomics and transcriptomics analyses to investigate whether there were detectable changes in the host that we could attribute to *C. neoformans* infection.

A

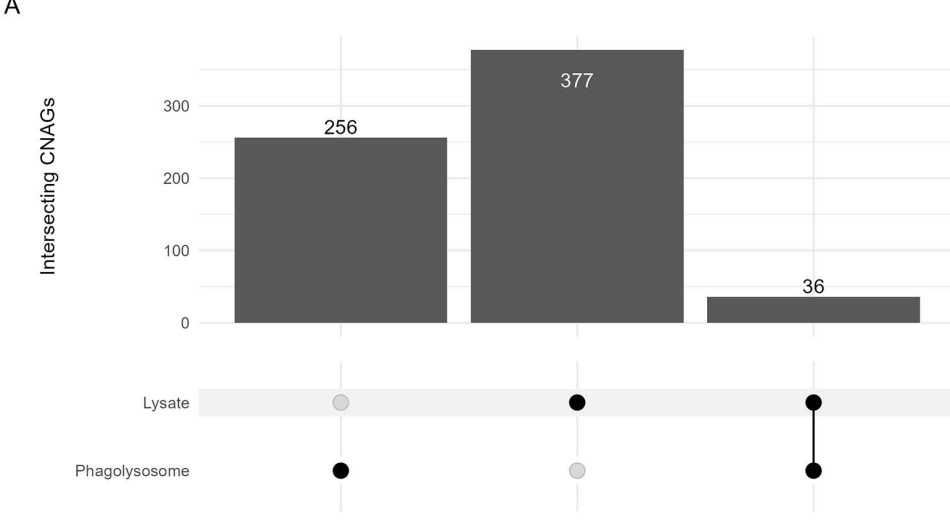

B

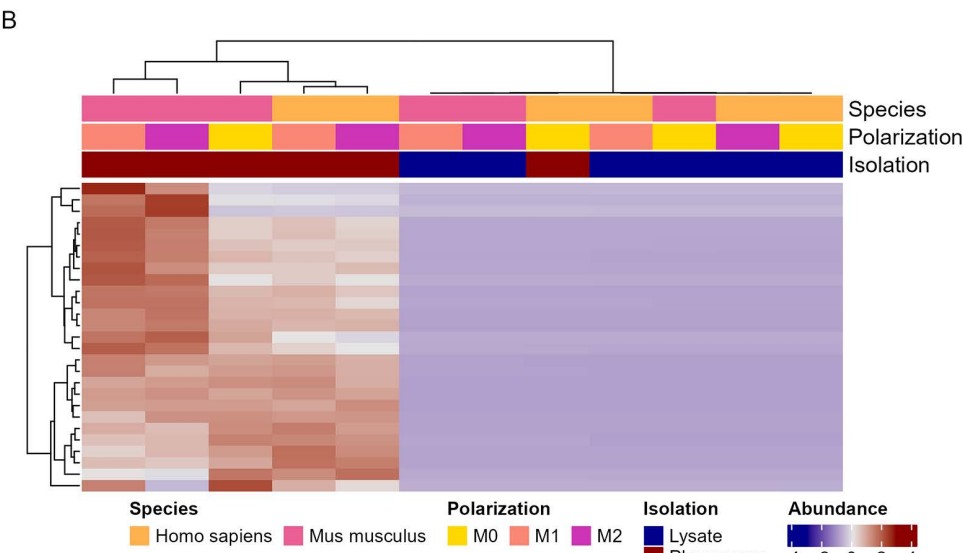

**Fig 4. Analysis of the 36 *C. neoformans* proteins detected in both whole cell lysate and isolated phagosomes of variously polarized mouse and human and compiled into quantitative protein data. A.** Upset plot depicting unique and overlapping detected proteins from each experiment. **B.** Abundances of each detected protein, normalized by row. Each condition is comprised of three biological replicates, each in technical triplicate.

We focused on proteins identified in the isolated phagosome samples to further investigate fungal-mammalian interactions. We identified 7933 proteins in human isolated phagolysosomes and 8038 in mouse isolated phagolysosomes (Fig 7 and S15–S20 Tables). In both species, GO and KEGG enrichment analysis revealed highly enriched pathways relating to metabolic, GTPase, and restructuring activities as well as anti-intracellular pathogen and DNA repair responses (S6 Fig). Notably, markers of cell stress and death are also enriched in both lists.

Of the identified proteins in both lists, we were able to identify 3,579 orthologs present in both mouse and human samples (Fig 8A and S21–S23 Tables). A set of 15 of these genes were detected in significantly different abundances in M1 polarization compared to M2 in both species and in the same direction (Fig 8B and S24–S26 Tables). As expected,

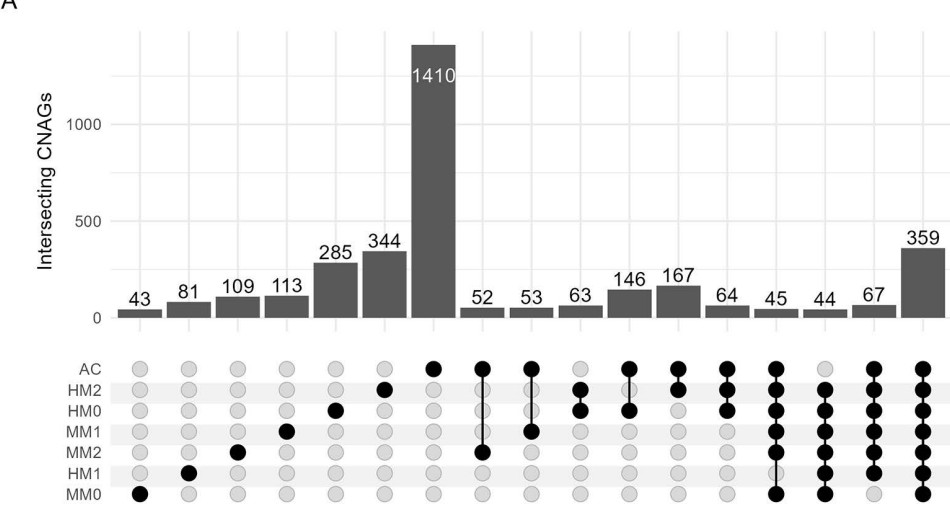

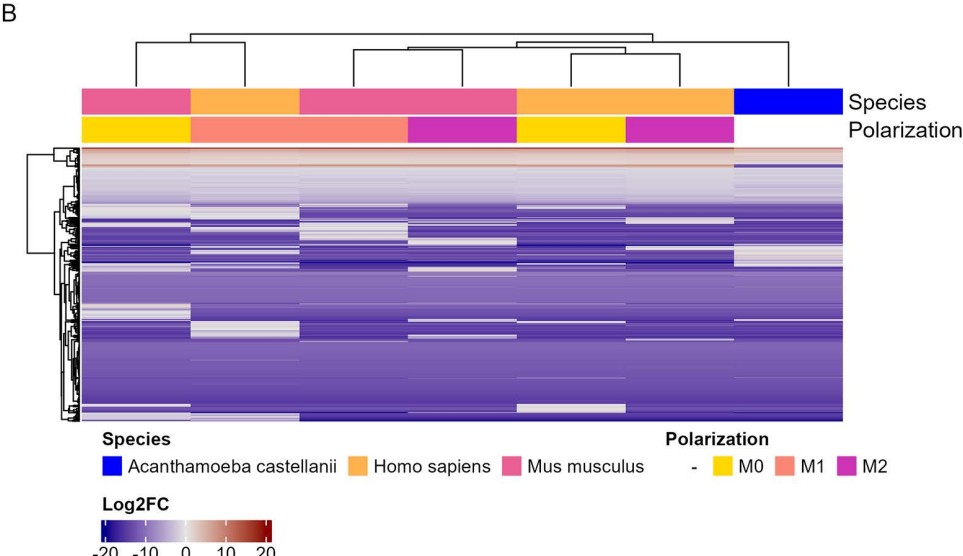

**Fig 5. Transcriptomic analysis of *C. neoformans* ingested by *A. castellanii* as well as human and mouse macrophages of various polarizations compared to preculture grown in nutrient rich media. A.** Upset plot depicting significant overlapping (*P* < 0.05) differentially expressed genes of each sample. Data is truncated to intersections of at least 40, full data is in supplementary materials. **B**. Log2FC map of the 359 genes which were differentially expressed in all samples. Each condition is comprised of three biological replicates, each in technical triplicate.

ontology enrichment for the entire list of orthologues resembles the two parental lists with a heavy emphasis on structural and metabolic changes (S7A and S7C Fig). When paring this list down to compare proteins more abundant in high stress (M1) phagocytes, we observed similar ontologies with a heavy focus on inflammation response (S7B and S7D Fig).

## Discussion

The global public health burden of fungal pathogens has been proposed to increase as warming global temperatures select for heat tolerant strains of environmental fungi, leading to new emerging fungal pathogens [29]. We are potentially already seeing the effects of this phenomenon in the emergence of pathogens like *C. auris* [30]. Those fungal species

A

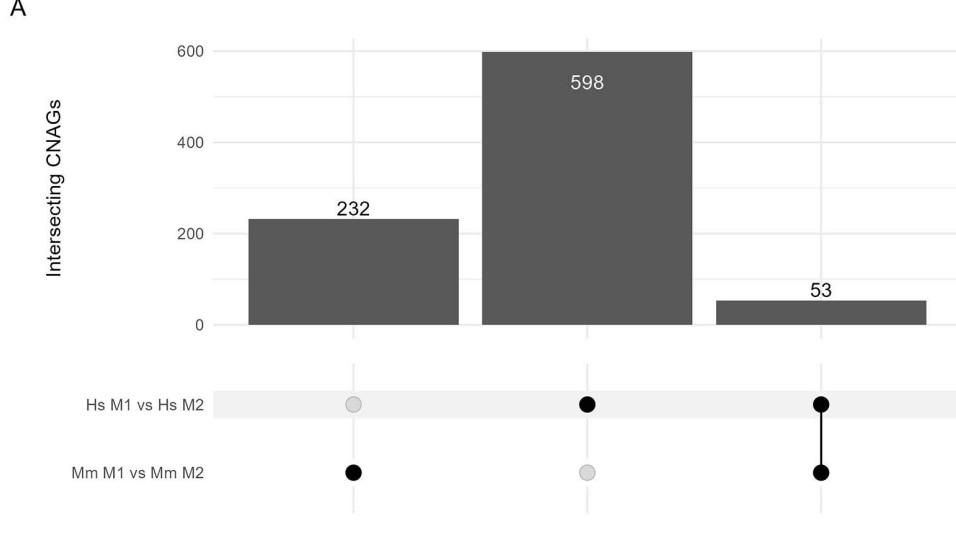

B

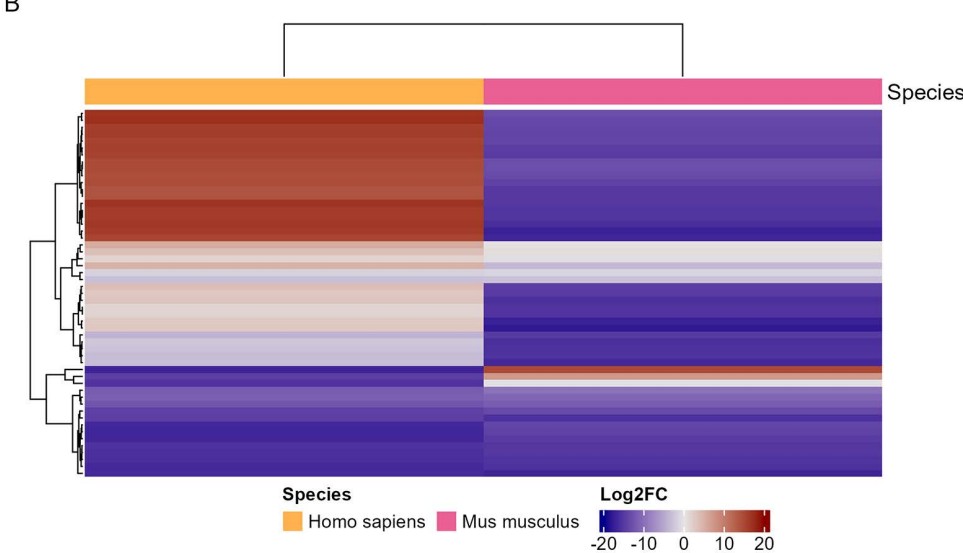

**Fig 6. Transcriptomic analysis of mouse derived macrophages and human circulating monocytes of M1 polarization compared to M2 polarization. A.** Upset plot of DEGs and overlap between the two sets. **B.** Log2FC of the 53 common DEGs between both groups. **C.** GO enrichment analysis of the 53 common DEGs. Enrichment and scoring are based on Fishers' exact test on the weight01 algorithm. Each condition is comprised of three biological replicates, each in technical triplicate.

preyed on by phagocytic organisms in the environment could have a head start in the cellular arms race between host and pathogen as they have had millions of years to adapt to amoeboid hosts, which resemble macrophage-like cells in form and function. To investigate the fungal response to ingestion and the aspects that are conserved between amoeba and mammalian hosts, we employed a combination of proteomic and transcriptomic analysis of host and fungal cells during *in vitro* infections.

In this study we expanded the scope of our recent investigation of *C. neoformans* proteins secreted after phagocytic ingestion [9]. We sought to compare cryptococcal proteins in host cell lysates after ingestion by different host species as well as polarization states of phagocytic cells. M1 polarized populations of mouse BMDMs and human PBMCs were

**Table 1. Significant genes and proteins found in our datasets compared to known phenotypes from established literature.**

| CNAG | Gene ID | Description | Proteomics | | Transcriptomics | Known Effect[b] |
|---|---|---|---|---|---|---|
| | | | Phagosome | Lysate | | |
| CNAG_04388 | sod2 | mitochondrial manganese superoxide dismutase | X[a] | X[a] | X[a] | ↓V [42] |
| CNAG_00130 | rck2 | AMK/CAMK1/CAMK1-RCK protein kinase | | X | X | ↑U [43] \| ↓AF [43] |
| CNAG_01019 | sod1 | superoxide dismutase [Cu-Zn] | | X | X | ↓V [42] |
| CNAG_05771 | tel1 | serine/threonine-protein kinase TEL1 | | X | | ↓Dn [44] |
| CNAG_03894 | pdr802 | Putative Zn2-Cys6 zinc-finger transcription factor | | | X | ↓V [45, 46] \| ↓AF [45] |
| CNAG_05431 | rim101 | pH-response transcription factor pacC/RIM101 | | | X | ↑V [47] \| ↑U [48] |
| CNAG_02801 | trx1 | thioredoxin | | | X | ↓V [49] |
| CNAG_00575 | cat3 | Catalase 3 | | | X | ↓D [21] |
| CNAG_01355 | trxH | Thioredoxin, variant | | | X | ↓D [21] |
| CNAG_00162 | aox1 | alternative oxidase, mitochondrial | | | X | ↓V [50] |
| CNAG_06287 | gpx2 | Glutathione peroxidase 2 | | | X | ↑D [21] |

[a]X denotes protein identified in proteomics experiments or gene marked as significantly differentially expressed in transcriptomics experiments.

[b]Known effects of knockout mutants for each respective gene during the interaction of C. neoformans and macrophages. D = Dragotcytosis (macrophage to macrophage transfer) frequency, V = virulence, Dn = resistance to DNA damage, U = urease activity, AF = resistance to Fluconazole or Amphotericin B.

**Table 2. Significant genes and proteins found in our datasets, without published phenotypes, that we found to have an effect on pathogenesis in the form of Dragotcytosis frequency when knocked out.**

| CNAG | Gene ID | Description | Proteomics | | Transcriptomics | Effect[a] |
|---|---|---|---|---|---|---|
| | | | Phagosome | Lysate | | |
| CNAG_04744 | m6pi | mannose-6-phosphate isomerase | | X | X | ↓D |
| CNAG_00130 | rck2 | AMK/CAMK1/CAMK1-RCK protein kinase | | X | X | ↓D |
| CNAG_05771 | tel1 | serine/threonine-protein kinase TEL1 | | X | | ↓D |
| CNAG_01722 | vsp13 | vacuolar protein sorting-associated protein vps13 | | | X | ↑D |
| CNAG_06332 | sas3 | putative histone acetyltransferase | | | X | ↑D |
| CNAG_02103 | | Unspecified | | | X | ↑D |
| CNAG_04117 | apaf1 | putative APAF1 ortholog | | | X | ↑D |
| CNAG_01048 | | Unspecified | | | X | ↑D |

[a]D = Dragotcytosis (macrophage to macrophage transfer) frequency.

chosen to compare host species as they are best suited for dealing with fungal pathogens and seem the most appropriate comparison to amoeba phagolysosomes [21]. When comparing cryptococcal proteins secreted into host cells during infection, we found that the largest differences were between amoeba and mammalian hosts, which is perhaps not surprising considering amoeba diverged from animals in the eukaryotic branch of life even earlier than yeast [31]. While we expect overlap due to the Amoeba predation hypothesis, each species phagosomal environment is likely to differ as well as *C. neoformans* response to specific differing stimuli. However, there was also a subset of proteins conserved across all three species that we propose reflects a common fungal core response against predatory ingestion, as the process of phagocytosis and phagosome formation are highly conserved. Analysis of cryptococcal proteins revealed a high prevalence of those involved in stress responses, metabolic changes, and anabolic processes, which we surmised is the result of fungal cell adaptation as the yeast transitioned from the extracellular space to the nutrient deprived environment of the phagolysosome.

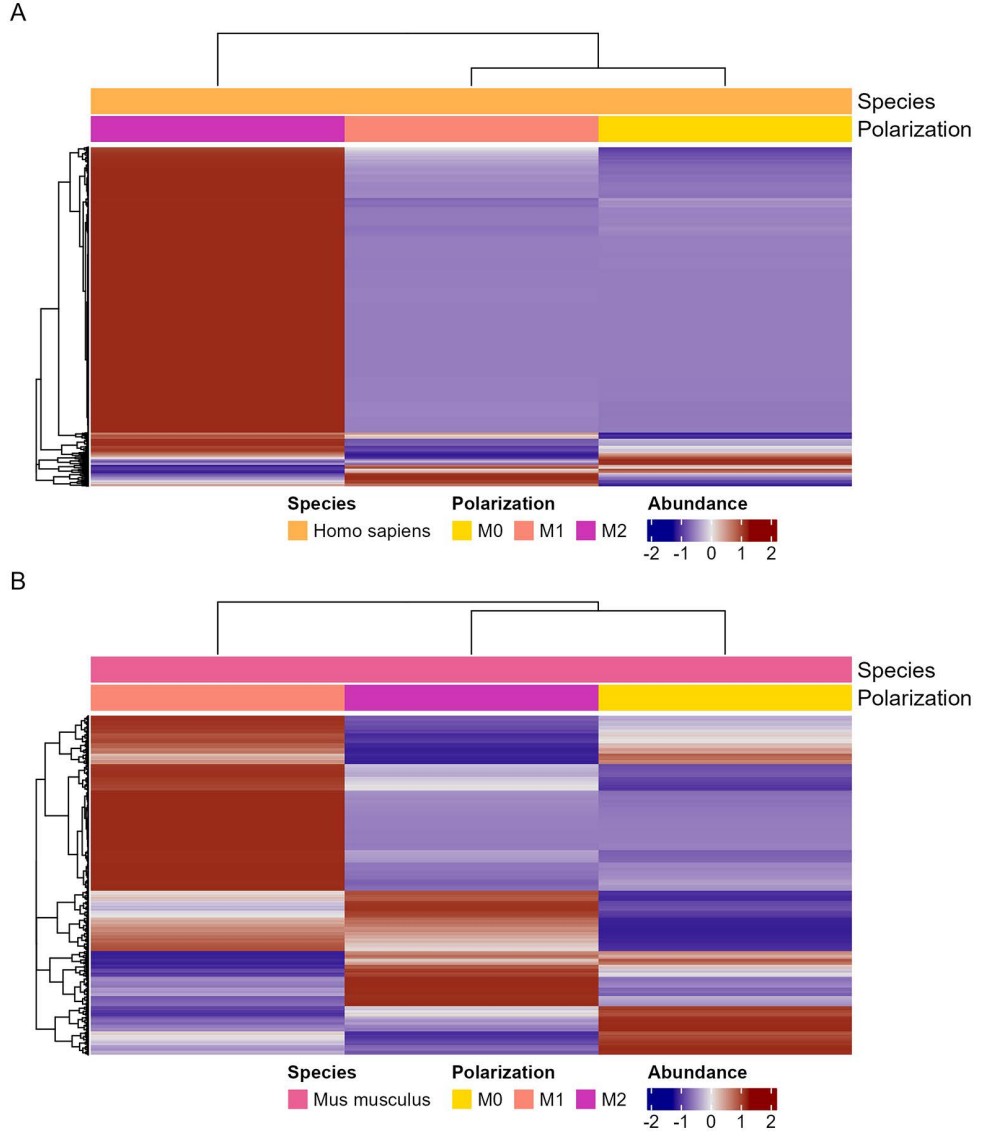

**Fig 7. Proteomic analysis of host proteins isolated from phagosomes containing *C. neoformans* and compiled into quantitative protein data.**
**A.** Abundances of identified proteins from isolated human phagolysosomes. **B.** Abundances of identified proteins from isolated mouse phagolysosomes. **C.** GO enrichment analysis of the identified human proteins. **D.** GO enrichment analysis of the identified human proteins. Enrichment and scoring are based on Fishers' exact test on the weight01 algorithm. **E.** KEGG pathway analysis of the identified human proteins based on Fishers exact test. **F.** KEGG pathway analysis of the identified mouse proteins based on Fishers exact test. Each condition is comprised of three biological replicates, each in technical triplicate.

M1 macrophages are inhospitable to *C. neoformans* while M2 macrophages are permissive, resulting in different outcomes to ingestion [21]. We compared proteins isolated from whole cell lysate of murine BMDMs and human PBMCs polarized to either M0, M1, or M2. Again, we found the most significant differences in cryptococcal protein expression when comparing different species. Surprisingly, there is notable consistency in the suite of secreted proteins between polarization states within the same host species, which may reflect a fungal response to ingestion irrespective of the stresses experienced by fungal cells. There was more variability within the human PBMCs than in the murine BMDMs,

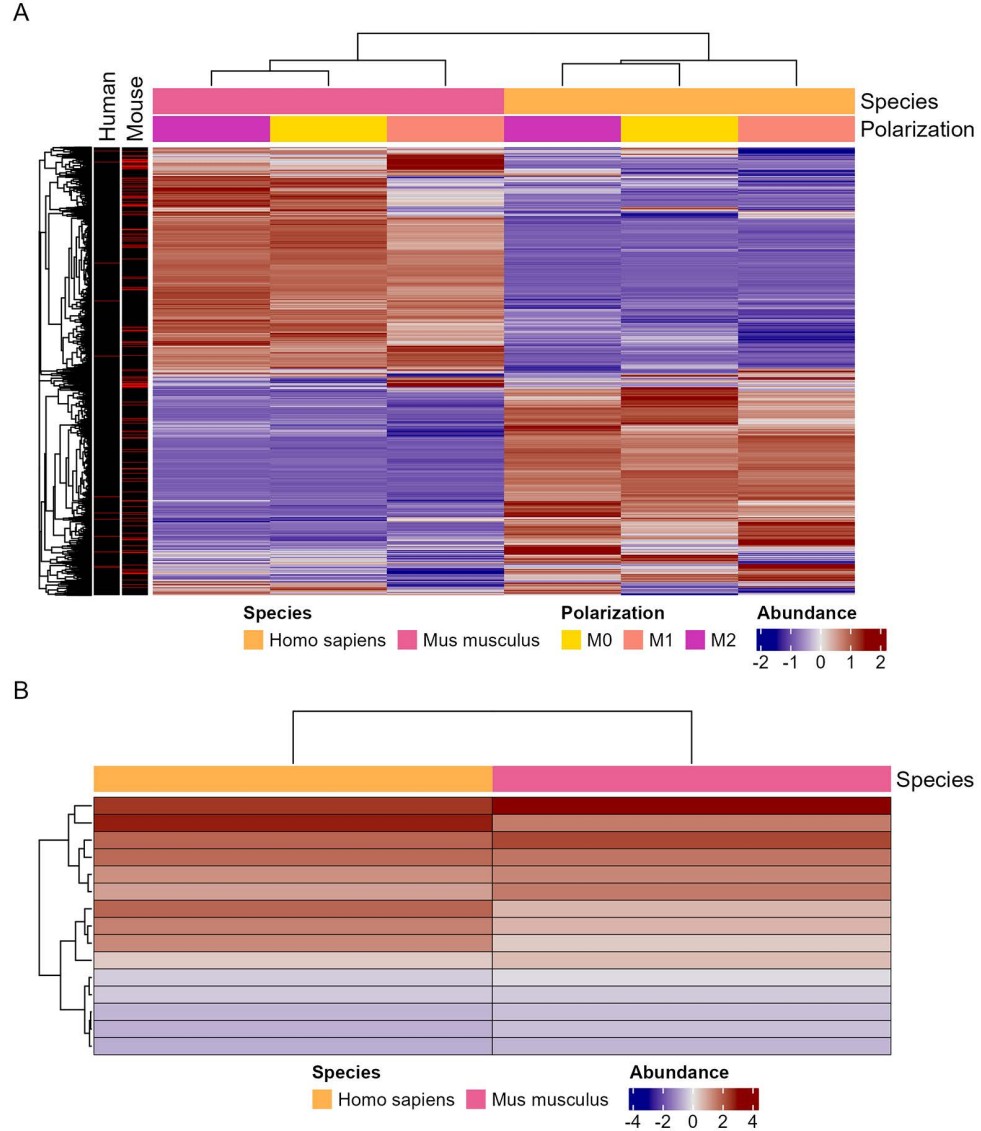

**Fig 8. Proteomic analysis of overlapping host protein orthologs isolated from phagosomes containing *C. neoformans* and compiled into quantitative protein data. A.** Abundances of known orthologs identified in both human and mouse phagolysosomes. Annotation depicts proteins identified in significantly different abundances when comparing M1 to M2 phagocytes of the same species. **B.** Abundances of 15 orthologs with significant abundance differences in both human and mouse M1 compared to M2 phagocytes. **C.** GO enrichment analysis of the entire list of orthologs. **D.** GO enrichment analysis of the 15 shared significant orthologs. Enrichment and scoring are based on Fishers' exact test on the weight01 algorithm. **E.** KEGG pathway analysis of the entire list of orthologs based on Fishers exact test. **F.** KEGG pathway analysis of the 15 shared significant orthologs based on Fishers exact test. Each condition is comprised of three biological replicates, each in technical triplicate.

with almost identical responses in murine M0 and murine M2 populations. Part of this effect may be due to the nature of the human PBMCs, which are subject to significantly more variability in donor, health, activation status, etc. compared to the mouse primary derived bone marrow differentiated macrophages. Furthermore, murine macrophages differ from human macrophages in that these can produce copious amounts of nitric oxide, which is toxic to *C. neoformans* [32]. We observed similar metabolic responses, such as the shift to gluconeogenesis, but narrowing our focus to the mammalian host also showed changes to protein generation and trafficking.

A disadvantage of the cell-lysis method is that we are attempting to parse out changes in *C. neoformans* proteins in host cell lysates, where fungal proteins represent an extremely small proportion of the total proteins found in the lysate. To address this limitation, and to enrich for cryptococcal proteins, we adapted a recently published protocol to isolate *Aspergillus fumigatus*-containing phagosomes and report the first isolation of the cryptococcal phagosome [18]. We consider this a significant technical achievement given that our prior efforts to recover cryptococcal phagolysosomes over many years were unsuccessful, which we attributed to the vulnerability of such large structures to shear forces [9]. Isolating phagolysosomes for proteomic analysis greatly reduced the ratio of host to fungal peptides and allowed the identification of a larger set of fungal proteins. Focusing on cryptococcal phagosomal proteomes also narrows the scope of the analysis to just the phagolysosome, where most of the relevant host-fungal interactions are likely to occur since it is in that compartment where fungal cells are either killed or proliferate to overwhelm the host cell. Unfortunately, this method failed with amoebae as we were unable to isolate phagolysosomes from amoeba hosts, which we attribute to their resistance to inhibition at 4 °C and ability to migrate then encyst during overnight incubation. We found similar results as observed with whole cell lysate in that most differences were between host species, with barely any differences in abundance between polarization states of the same host species.

Additionally, we identified subsets of proteins that appeared only in either the whole cell lysate or isolated phagolysosomal samples. This could represent a subset of proteins that are explicitly secreted into the host cell cytosol rather than just released by *C. neoformans* into whichever extracellular milieu it finds itself in. We note that cryptococcal cells in macrophages release copious amounts of vesicles into the host cell cytoplasm that contain capsular polysaccharide and could include a different set of cryptococcal proteins [15]. It is also possible that leaky phagosomes, caused by *C. neoformans* infection, result in passive diffusion of fungal proteins into the host cell [28]. Determining whether fungal proteins are actively transported through the phagolysosomal membrane would require extensive fluorescence-activated cell sorting of non-leaky phagosomes followed by isolation and mass spectroscopy, which is beyond the scope of this work.

Concurrently, we investigated the transcriptional response of ingested *C. neoformans* using RNAseq. We first compared *C. neoformans* isolated from all hosts and polarization states to *C. neoformans* grown in nutrient rich YPD media. We found 359 genes which were significantly differentially regulated across all hosts and polarization states compared to YPD growth. This list included several known genes important to capsule independent anti-phagocytic mechanisms [33]. Also included is *Blp1,* known to be essential to *C. neoformans* virulence and phagocytic defense which has also been shown to be upregulated during *C. neoformans* lung infection of *Mus musculus* and *Macaca fascicularis [*34]. Overall, these 359 common DEGs are also associated with metabolic changes and response to stress. Finding these genes consistently across different phagocytic environments and similar ontologies enriched by both proteomics and transcriptomics also places our results within the framework of established knowledge obtained by prior studies, reassuring us in the validity of our findings. Furthermore, we were able to link several additional genes to virulence phenotypes associated with Dragotcytosis. Several candidates completely inhibited Dragotcytosis when knocked out. While the rarity of these events across both species makes it difficult to identify statistical significance without extreme sample sizes, we still find it intriguing that no transfer events were observed, something we have not seen in any other knockouts. Taken together, this data affirms our confidence that our remaining list of uncharacterized genes and proteins will contain new and important molecular players in *C. neoformans* pathogenesis. However, a significant limitation in our analysis is that the *C. neoformans* genome is not fully annotated, such that less than half of the reference genes are associated with at least one GO term. Thus, our analysis is currently limited but the dataset that we have generated should be suitable for future analysis and expect that additional ontology terms and pathways of interest will be revealed with annotation is more complete.

Analysis of human and mouse proteins from isolated *C. neoformans*-containing phagolysosomes shows mainly metabolism and proteasome related pathway enrichment along with inflammation and pathogen response. Apoptotic effectors and inflammation associated caspases were detected in both mice and human phagocytes with significant enrichment for apoptotic pathways. Along with the presence of SSB and Hsp90, these data emphasize the presence of expected

                                   

biomarkers, further confirming the presence of cell damage and supporting the accuracy of our findings [35]. Our list of knowns coincides with established literature, and our list of unknowns has yielded potential targets to disrupt Dragotcytosis pathways. When analyzing host proteins isolated from *C. neoformans*-containing phagolysosomes from M1 compared to M2 phagocytes, we unsurprisingly found the strongest and most consistent signals from a suite of innate immune response and inflammation effectors one would expect of M1 polarization.

We previously showed that when dealing with a broad and unknown threat, macrophages will hedge bets in terms of antimicrobial defenses [36]. In the environment *C. neoformans* faces varied environmental predators and would presumably benefit from a similarly broad strategy. Since that there are thousands of amoeba species that are potential fungal predators, and that fungal cells do not know the identity of their predators, it follows that their survival strategy must be non-predator species specific, which implies targeting common metabolic pathways [37]. Given that eukaryotic cell metabolism is tightly regulated, disruption of metabolic activities suggests a common target in phagocytic cells [38]. Thus, we hypothesize that the long list of metabolic enzymes reported here that are released into the host cell in these experiments could represent a general defense/offense mechanism, whereby *C. neoformans* shunts enzymes into the host cell that can disrupt the delicate homeostasis of its metabolic pathways. According to this hypothesis, fungal-mediated metabolic disruption would affect host cell homeostasis, impair its fungicidal activity and increase the likelihood of fungal cell survival. We suspect that EVs could be one such delivery mechanism. While the EV field is still relatively young, our data shows similarities with other lists of both proteins and mRNAs contained within *C. neoformans* EVs [39]. Interestingly, we see higher consistency between our list of identified proteins and their respective mRNAs previously found in cryptococcal EVs [40,41].

In summary, the most remarkable finding in this study was the similarity in cryptococcal proteins analyzed across lysates from amoeba and phagocytes from mice and humans. Although there were also significant differences in the *C. neoformans* response to ingestion by three different host species the overall commonalities suggest that when cryptococci experience the stress of ingestion and phagosomal residence they produce a suite of common proteins. This finding is consistent with, and supportive of, the hypothesis that the capacity for mammalian virulence emerged in part from selection of virulence traits that allow survival in the environment when fungal cells are subject to prey by amoeboid cells. Given tens of thousands of protists with capacity to prey on *C. neoformans* [37], this fungus is under tremendous pressure to develop effective defense mechanisms against very different phagocytic predators. Since *C. neoformans* cells do not know the identity of phagocytic predators, it must have a common approach to deal with such threats. We propose that the shared cryptococcal proteins identified in amoeba, mouse BMDMs, and human PBMCs constitute a toolkit for intracellular survival that is effective against phylogenetically distant phagocytic predators, which provides a potential explanation for the mechanism that makes *C. neoformans* a generalist type of pathogen. Given the technical limitations of our experiments noted above we suspect that the proteins identified in this study represent only a partial set of such a toolkit. Taken together, the resulting protein sets observed in these two mammalian phagocytic cells established a foundation for us* to investigate both the conserved *C. neoformans* response to ingestion as well as species unique elements of this response. We hypothesize these unique elements may be ideal targets for understanding why differences exist in pathogenesis between hosts. For example, Dragotcytosis (transfer of *C. neoformans* between hosts cells) has not been directly observed for amoeba, while Vomocytosis (non-lytic exocytosis of *C. neoformans*) has, and Dragotcytosis and Vomocytosis occur in different frequencies depending on host macrophage polarization [21]. Perhaps the machinery required for this interaction can be identified by their high abundance in M1 macrophages compared to M2 macrophages and amoeba and our study provides a clear point of departure for future studies.

## Supporting information

**S1 Fig. GO and KEGG enrichment analysis of fungal proteins isolated from host cell lysates. A.** GO enrichment analysis of the entire list of 359 peptides identified across all three host species of cell lysates. Enrichment and scoring are based on Fishers' exact test on the weight01 algorithm. **B.** GO enrichment analysis of the entire list of proteins identified

across polarizations of mouse and human phagocytes. Enrichment and scoring are based on Fishers' exact test on the weight01 algorithm. Results are consolidated to parental terms. **C.** KEGG pathway analysis of proteins identified across host species based on Wilcoxon rank sum test. MF: Molecular Function, BP: Biological Process, CC: Cellular Component ontologies.
(TIF)

**S2 Fig. Phagosomes containing GFP-actin expressing *C. neoformans* stained with human anti-V-ATPase. A)** 40x isolated phagosomes. Phase contrast, GFP-Actin *C. neoformans*, stained with 1:100 anti-V-ATPase E1 polyclonal anti-body (PA5–29899) and counterstained with Goat Anti-Rabbit IgG H&L conjugated to a Texas Red fluorophore (ab6719) **B)** 63X isolated phagosome.
(TIF)

**S3 Fig. GO and KEGG enrichment analysis of proteins isolated from yeast-containing phagolysosomes and 42 *C. neoformans* proteins detected in both whole cell lysate and isolated phagosomes. A.** GO enrichment analysis of the entire list of 681 proteins identified across all three host species of isolated phagolysosomes. Enrichment and scoring are based on Fishers' exact test on the weight01 algorithm. **B.** GO enrichment analysis of the entire list of 42 proteins identified in both phagolysosome and lysate samples. Enrichment and scoring are based on Fishers' exact test on the weight01 algorithm. Results are consolidated to parental terms. **C.** KEGG pathway analysis of proteins from isolated phagolysosomes based on Wilcoxon rank sum test. **D.** KEGG pathway analysis of proteins identified in both phagolysosomes and lysates based on Wilcoxon rank sum test. MF: Molecular Function, BP: Biological Process, CC: Cellular Component ontologies.
(TIFF)

**S4 Fig. GO enrichment analysis of the 359 common DEGs across hosts and polarization states.** Enrichment and scoring are based on Fishers' exact test on the weight01 algorithm. Results are consolidated to parental terms. MF: Molecular Function, BP: Biological Process, CC: Cellular Component ontologies.
(TIFF)

**S5 Fig. Genes of interest identified in this study which display significant phenotypes when knocked out.** Drag-otcytosis frequency is modulated during *in vitro* BMDM infection of several genes identified as significant in this list but largely absent from wider literature. Error bars represent 95% CI of mean frequency. Significance was determined via test of equal proportions compared to wild-type KN99α and corrected for multiple hypothesis via Benjamini-Hochberg. *, **, ***, and **** denote P < 0.05, 0.01, 0.001, and 0.0001, respectively.
(JPEG)

**S6 Fig. GO and KEGG enrichment analysis of host proteins identified from yeast-containing phagolysosomes. A.** GO enrichment analysis of the entire list of 7933 human proteins identified from isolated phagolysosomes. Enrichment and scoring are based on Fishers' exact test on the weight01 algorithm. **B.** GO enrichment analysis of the entire list of 8038 mouse proteins identified from isolated phagolysosomes. Enrichment and scoring are based on Fishers' exact test on the weight01 algorithm. Results are consolidated to parental terms. **C.** KEGG pathway analysis of human proteins from isolated phagolysosomes based on Wilcoxon rank sum test. **D.** KEGG pathway analysis of mouse proteins identified in both phagolysosomes and lysates based on Wilcoxon rank sum test. Both GO and KEGG analyses are limited to the 10 most highly represented groups for visualization with complete results in the supplement. MF: Molecular Function, BP: Biological Process, CC: Cellular Component ontologies.
(TIFF)

**S7 Fig. GO and KEGG enrichment analysis of orthologous host proteins identified from yeast-containing phagolysosomes. A.** GO enrichment analysis of the entire list of 3279 proteins identified from isolated

phagolysosomes in both species that are known orthologues. Enrichment and scoring are based on Fishers' exact test on the weight01 algorithm. **B.** GO enrichment analysis of the 15 orthologues detected in different abundances between M1 and M2 polarization in both species. Enrichment and scoring are based on Fishers' exact test on the weight01 algorithm. Results are consolidated to parental terms. **C.** KEGG pathway analysis of the entire list of known orthologues based on Wilcoxon rank sum test. **D.** KEGG pathway analysis of the 15 commonly different proteins based on Wilcoxon rank sum test. Both GO and KEGG analyses are limited to the 10 most highly represented groups for visualization with complete results in the supplement. MF: Molecular Function, BP: Biological Process, CC: Cellular Component ontologies.

(TIFF)

**S1 Table. Proteomics of C. neoformans infected whole cell lysates of BMDMs, PBMCs, and Amoebas.**
(XLSX)

**S2 Table. GO enrichment analysis of C. neoformans infected whole cell lysates of BMDMs, PBMCs, and Amoebas.**
(XLSX)

**S3 Table. KEGG enrichment analysis of C. neoformans infected whole cell lysates of BMDMs, PBMCs, and Amoebas.**
(XLSX)

**S4 Table. Proteomics of isolated C. neoformans-containing phagosomes.**
(XLSX)

**S5 Table. GO enrichment analysis of C. neoformans infected whole cell lysates of M0, M1, and M2 polarized BMDMs and PBMCs.**
(XLSX)

**S6 Table. Proteomics of isolated C. neoformans-containing phagosomes.**
(XLSX)

**S7 Table. GO enrichment analysis of isolated C. neoformans-containing phagolysosomes of M0, M1, and M2 polarized BMDMs and PBMCs.**
(XLSX)

**S8 Table. KEGG enrichment analysis of isolated C. neoformans-containing phagolysosomes of M0, M1, and M2 polarized BMDMs and PBMCs.**
(XLSX)

**S9 Table. GO enrichment analysis of common proteins among isolated phagosome samples.**
(XLSX)

**S10 Table. KEGG enrichment analysis of common proteins among isolated phagosome samples.**
(XLSX)

**S11 Table. 359 significant C. neoformans DEGs after ingestion by BMDMs, PBMCs, and Amoebas compared to stationary YPD cultures.**
(XLSX)

**S12 Table. GO enrichment analysis of 359 significant C. neoformans DEGs after ingestion by BMDMs, PBMCs, and Amoebas compared to stationary YPD cultures.**
(XLSX)

**S13 Table. 42 significant C. neoformans DEGs after ingestion by BMDMs and PBMCs M2 vs M1.**
(XLSX)

**S14 Table. GO enrichment analysis of commonly significant C. neoformans DEGs after ingestion by BMDMs and PBMCs M2 vs M1.**
(XLSX)

**S15 Table. Human proteins identified from isolated C. neoformans-containing phagolysosomes.**
(XLSX)

**S16 Table. GO enrichment analysis of human proteins identified from isolated C. neoformans-containing phagolysosomes.**
(XLSX)

**S17 Table. KEGG enrichment analysis of human proteins identified from isolated C. neoformans-containing phagolysosomes.**
(XLSX)

**S18 Table. Mouse proteins identified from isolated C. neoformans-containing phagolysosomes.**
(XLSX)

**S19 Table. GO enrichment analysis of mouse proteins identified from isolated C. neoformans-containing phagolysosomes.**
(XLSX)

**S20 Table. KEGG enrichment analysis of mouse proteins identified from isolated C. neoformans-containing phagolysosomes.**
(XLSX)

**S21 Table. Orthologous mouse and human proteins identified in C. neoformans containing phagolysosomes.**
(XLSX)

**S22 Table. GO enrichment analysis of orthologous mouse and human proteins identified in C. neoformans containing phagolysosomes.**
(XLSX)

**S23 Table. KEGG Enrichment Analysis of Orthologous mouse and human proteins identified in C. neoformans containing phagolysosomes.**
(XLSX)

**S24 Table. Orthologous mouse and human proteins identified in C. neoformans containing phagolysosomes significant in both species M2 vs M1.**
(XLSX)

**S25 Table. GO enrichment analysis of orthologous mouse and human proteins identified in C. neoformans containing phagolysosomes significant in both species M2 vs M1.**
(XLSX)

**S26 Table. KEGG enrichment analysis of orthologous mouse and human proteins identified in C. neoformans containing phagolysosomes significant in both species M2 vs M1.**
(XLSX)

## Acknowledgments

We would like to acknowledge the hard work of Bob Cole, Bob O'Meally, and Jeremy Post of the Johns Hopkins University School of Medicine Mass Spectrometry and Proteomics Core as well as Georgia Stavrakis and the Andrea Cox lab for training us in human monocyte culture. We would also like to acknowledge VEuPathDB as none of this would have been possible without FungiDB and AmoebaDB.

## Author contributions

**Conceptualization:** Quigly Dragotakes.

**Data curation:** Quigly Dragotakes.

**Formal analysis:** Quigly Dragotakes, Ella Jacobs, Anne Jedlicka, Amanda Dziedzic.

**Funding acquisition:** Arturo Casadevall.

**Investigation:** Quigly Dragotakes, Ella Jacobs, Gracen Gerbig, Seth Greengo, Anne Jedlicka, Amanda Dziedzic.

**Methodology:** Quigly Dragotakes, Ella Jacobs, Gracen Gerbig, Seth Greengo, Anne Jedlicka, Amanda Dziedzic.

**Visualization:** Quigly Dragotakes, Ella Jacobs, Anne Jedlicka, Amanda Dziedzic.

**Writing – original draft:** Quigly Dragotakes, Ella Jacobs, Anne Jedlicka, Amanda Dziedzic, Arturo Casadevall.

**Writing – review & editing:** Quigly Dragotakes, Ella Jacobs, Gracen Gerbig, Seth Greengo, Anne Jedlicka, Amanda Dziedzic, Arturo Casadevall.

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
