## [Decision Letter · Decision Letter 0]

29 Jul 2025

A Conserved Enzymatic Toolkit Targeting Host Cell Metabolism is Associated with Cryptococcus neoformans Intracellular Survival in Protozoal and Mammalian Phagocytic Cells

PLOS Pathogens

Dear Dr. Dragotakes,

Please submit your revised manuscript within 60 days Sep 27 2025 11:59PM. If you will need more time than this to complete your revisions, please reply to this message or contact the journal office at plospathogens@plos.org. Please include the following items when submitting your revised manuscript:

We look forward to receiving your revised manuscript.

Kind regards,

Felipe H Santiago-Tirado, PhD

Guest Editor

PLOS Pathogens

Michal Olszewski

Section Editor

Editor-in-Chief

PLOS Pathogens

Michael Malim

Editor-in-Chief

PLOS Pathogens

orcid.org/0000-0002-7699-2064

**Additional Editor Comments:**

Dear Dr. Dragotakes, I, as the three reviewers, agree that this is an interesting study that will be of general interest to the cryptococcal field as well to the general microbial pathogenesis community. However, the numerous stylistic concerns (regarding data presentation and description, and confusing methodology) will limit its impact. Moreover, all three of them raised questions about the validity of all the proteins identified: are they really phagosomal proteins? Do they have a physiologically-relevant effect on the host phagosome or the fungus intracellular niche? I believe the limitations of the study need to be clearly stated, especially why "dragotcytosis" was used to validate some of the hits rather than a more relevant assay like phagosomal pH or intracellular replication/survival. Still, we all agree these results will be of broad interest and useful to the field. I look forward to your resubmission. 

**Journal Requirements:**

At this stage, the following Authors/Authors require contributions: Quigly Dragotakes, Ella Jacobs, Gracen Gerbig, Seth Greengo, Anne Jedlicka, Amanda Dziedzic, and Arturo Casadevall. Please ensure that the full contributions of each author are acknowledged in the "Add/Edit/Remove Authors" section of our submission form.

https://journals.plos.org/plospathogens/s/submission-guidelines#loc-parts-of-a-submission

3) Please insert an Ethics Statement at the beginning of your Methods section, under a subheading 'Ethics Statement'. It must include:

- The full name(s) of the Institutional Review Board(s) or Ethics Committee(s)

- The approval number(s), or a statement that approval was granted by the named board(s)

- A statement that formal consent was obtained (must state whether verbal/written) OR the reason consent was not obtained (e.g. anonymity). NOTE: If child participants, the statement must declare that formal consent was obtained from the parent/guardian.].

5) We notice that your supplementary Figures, and Tables are included in the manuscript file. Please remove them and upload them with the file type 'Supporting Information'. Please ensure that each Supporting Information file has a legend listed in the manuscript after the references list.

6) Please ensure that the funders and grant numbers match between the Financial Disclosure field and the Funding Information tab in your submission form. Note that the funders must be provided in the same order in both places as well.

**Reviewers' Comments:**

Reviewer's Responses to Questions

**Part I - Summary**

Reviewer #1: In this manuscript Dragotakes et al. have amassed a dataset of proteomics and transcriptomics aimed at further understanding the interaction of C. neoformans with phagocytes including macrophages (mouse BMDM), monocytes (human PBMC), and amoeba (Acanthamoeba castellanii). Specifically, the investigators are interested in how the Cryptococcal response to being eaten by these different types of cells is similar or different. The investigators conclude that while there are situation-specific differences, much of the Cryptococcal response to being eaten is shared in the models. Overall, it’s a straightforward premise and an interesting pile of data. I have only modest further questions/concerns:

Reviewer #2: Here, the authors successfully adapted a methodology for isolation of phagolysosomes that improved the identification of fungal cells after been in contact with host cells. This paper also analyses common routes employed by C. neoformans to defend themselves against predators and immune cells. In general, the paper is well written and organized. However, in order to improve its quality, some minor details should be addressed. For example, several results are described very lightly which forces the reader to go constantly to see the figure in question and halts the readability of this paper. Furthermore, the authors use several specific terminologies and abbreviations that need be explained to help the reader to understand the results. Lastly, there is no description in the figure caption related to statistical analysis and softwares used to create the images that limits the reproducibility of the results.

Reviewer #3: The manuscript by Dragotakes et al describes a proteomics and transcriptional response to phagocytosis of C. neoformans by mouse and human macrophages and Amoeba. The fascinating discovery is that the process of phagosome formation is highly similar among these species, suggesting that C. neoformans has evolved to trigger similar responses in phagocytes of vastly different origins. The paper is well-laid out and largely focuses on the results of multiple proteomics and transcriptional experiments. Overall, this is a large undertaking where these data will be valuable to the scientific community. My comments are largely minor in nature but should be addressed prior to consideration of publication.

**Part II – Major Issues: Key Experiments Required for Acceptance**

Reviewer #1: N/A

Reviewer #2: Line 145: Since the isolation of C. neoformans containing phagolysosomes is so problematic and the protocol was adapted from aspergillus to C. neoformans. Briefly describe changes for C. neoformans?

Line 258: Define Dragocytosis? This term is mentioned several times throughout the manuscript but is only briefly described at the end of the discussion. An earlier definition would improve the readability of the paper.

Line 268: Provide more details about number of proteins, examples of pathways, etc? Furthermore, please describe each panel individually.

Line 271-273: authors state that there is consistency of cryptotoccal peptides among species but there are clear differences in expression levels between Mm and Hs in Fig 1A and between humans and mouse and amoeba in Fig 1B.

Line 301: The 267 proteins differently abundant are in which condition? There are several comparisons in figure 3. Furthermore, please describe each panel individually.

Fig 1-3: Are the authors showing peptides or proteins? Although proteomics measures peptides, normally in bottom-up analysis like this, original proteins are traced back and showed. Likewise, if peptides, elaborate on the concept of expression levels for peptides?

Provide the meaning of abbreviations used, such as MM, Hs and Ac? Same about AR in X axis? Briefly describe which software and statistical analysis were used to generate the figures? Is the abundance relative ? If so, please state that.

Reviewer #3: 1. The time for phagocytosis is not clear. For the proteomics experiments, it suggests that C. neoformans and phagocytes were exposed for 2 hours. Was there a washout period of non-adhered fungi to ensure that phagosomes were of similar age (as determined in minutes/hours), or was C. neoformans present throughout? If the latter, then phagosomes could be 1 min to 120 minutes old. If the latter, how does one properly interpret the proteomics data.

2. For transcriptional analysis, the time of exposure of C. neoformans to phagocytes is not clear.

3. In all presented data where the authors’ emphasis is that the lists of proteins or upregulated genes are similar, the addition of Venn diagrams to volcano plots/ heat maps would be a great visual aid to make the case (as has been done in prior publications from the Casadevall lab).

**Part III – Minor Issues: Editorial and Data Presentation Modifications**

Reviewer #1: In figure 1B can we confirm that the color legend is correct? It looks like mouse and amoeba cluster together, while human is more different, but the rest of the data suggests mouse and human are more similar to each other.

It looks like the analyses were typically done with an n=3 (e.g. figure 1). Can we clarify whether these are technical or biological replicates? Other illustrations of the data can be less clear. Can we clarify in each case the n and whether and how samples were pooled for analysis throughout?

In the “purified” phagosome analyses of mouse/human proteins, are the investigators worried that the vast majority of these proteins are unlikely to be phagosome-associated? Clearly there are, for example, abundant of mitochondria. What is the evidence that these proteins are even enriched for phagosomal proteins? Or depleted of other organelles. There might be something to be done by comparison to the whole cell lysate data – albeit imperfect?

Supplementary Figure 1 – define BB, CC, MF: Molecular Function (MF), Biological Process (BP), and Cellular Component (CC).

Table1 and Table2 use differently formatted Gen IDs. Presumably all lower case is preferred.

Line 211: “The horizontal dendrogram shows the proteins in samples that clustered together.” Presumably this was copied from somewhere when a dendrogram was shown.

The text often refers to detecting peptides vs detecting proteins. I’m presuming that the processed data being analyzed throughout relates to peptide data that has already been compiled into quantitative protein data?

The method for polarizing BMDMs and making whole cell lysates is described, but not for the human monocytes? Presumably the same? Any modifications?

How were purified phagosomes processed for proteomics? Same lysis approach?

As a final note, I found that the story can be quite confusing to a reader trying to understand specifically when Cryptococcal proteins/genes are being assessed (usually) and when human/mouse/amoeba proteins or genes are being assessed. Any help with more overt and explicit labeling and text might be helpful.

Reviewer #2: Line 29-31: citation

Line 68: Elaborate on the explanation of “transfer events”?

Line 105: pH of acetate buffer?

Line 111: complete has an issue with a capital letter.

Line 111: pH of HEPES buffer?

Line 157. Was cell lysis performed without protease inhibitors? In their absence non-tryptic proteases could digest the proteins leading to less protein hits.

Line 412: which are “these”? Murine or human’s macrophages?

Supplemental Fig 1: Provide the meaning of abbreviations used, such as BB, CC and MF? Why are there different number of points on each row in the enrichment analysis? “Ontology is slightly cropped out of the image. Caption describes a D panel that does not exist in the image.

Supplemental Fig 3: several labels overlap.

Line 619: Ref 40: No information about authors, years, journal.

Reviewer #3: Minor point- when human peripheral blood is used, what is the status of IRB approval? Please state if approved or exempt.

PLOS authors have the option to publish the peer review history of their article (what does this mean? ). If published, this will include your full peer review and any attached files.

**Do you want your identity to be public for this peer review?** For information about this choice, including consent withdrawal, please see our Privacy Policy .

Reviewer #1: No

Reviewer #2: No

Reviewer #3: No

**Figure resubmission:**

**Reproducibility:**



---

## [Editor Report · Decision Letter 1]

4 Dec 2025

Dear Mr. Dragotakes,

We are pleased to inform you that your manuscript 'A Conserved Enzymatic Toolkit Targeting Host Cell Metabolism is Associated with Cryptococcus neoformans Intracellular Survival in Protozoal and Mammalian Phagocytic Cells' has been provisionally accepted for publication in PLOS Pathogens (see editor comments below).

Best regards,

Felipe H Santiago-Tirado, PhD

Guest Editor

PLOS Pathogens

Michal Olszewski

Section Editor

PLOS Pathogens

Sumita Bhaduri-McIntosh

Editor-in-Chief

PLOS Pathogens

orcid.org/0000-0003-2946-9497

Michael Malim

Editor-in-Chief

PLOS Pathogens

orcid.org/0000-0002-7699-2064

Dear Dr. Dragotakes, thank you for your resubmission. I found that you addressed all of the main reviewer concerns, including a better explanation of the methods/protocols and improved data presentation, thus I am pleased to editorially accept your manuscript. Please follow up with the staff regarding the next steps, and please pay attention during the proofs stage, as i found 2 typos (there might be more). Thank you for your contribution to the field.
---

## [Editor Report · Acceptance letter]

Dear Mr. Dragotakes,

We are delighted to inform you that your manuscript, "A Conserved Enzymatic Toolkit Targeting Host Cell Metabolism is Associated with *Cryptococcus neoformans* Intracellular Survival in Protozoal and Mammalian Phagocytic Cells," has been formally accepted for publication in PLOS Pathogens.

Best regards,

Sumita Bhaduri-McIntosh

Editor-in-Chief

PLOS Pathogens

orcid.org/0000-0003-2946-9497

Michael Malim

Editor-in-Chief

PLOS Pathogens

orcid.org/0000-0002-7699-2064